# GRU-SVM Based Threat Detection in Cognitive Radio Network

**DOI:** 10.3390/s23031326

**Published:** 2023-01-24

**Authors:** Evelyn Ezhilarasi I, J Christopher Clement

**Affiliations:** School of Electronics Engineering, Vellore Institute of Technology, Vellore 632014, Tamil Nadu, India

**Keywords:** spectrum sensing, gated recurrent unit, cognitive radio network, support vector machine, malicious users

## Abstract

Cognitive radio networks are vulnerable to numerous threats during spectrum sensing. Different approaches can be used to lessen these attacks as the malicious users degrade the performance of the network. The cutting-edge technologies of machine learning and deep learning step into cognitive radio networks (CRN) to detect network problems. Several studies have been conducted utilising various deep learning and machine learning methods. However, only a small number of analyses have used gated recurrent units (GRU), and that too in software defined networks, but these are seldom used in CRN. In this paper, we used GRU in CRN to train and test the dataset of spectrum sensing results. One of the deep learning models with less complexity and more effectiveness for small datasets is GRU, the lightest variant of the LSTM. The support vector machine (SVM) classifier is employed in this study’s output layer to distinguish between authorised users and malicious users in cognitive radio network. The novelty of this paper is the application of combined models of GRU and SVM in cognitive radio networks. A high testing accuracy of 82.45%, training accuracy of 80.99% and detection probability of 1 is achieved at 65 epochs in this proposed work.

## 1. Introduction

Intrusion attacks have been identified in software defined networks using deep learning models [1]. In [2], a deep recurrent neural network (RNN) was used to detect attacks in sofware defined networks. Based on certain deep learning techniques, attackers attacking the network have been identified and detected in [3]. Distributed denial of service attacks were addressed in [4] using LSTM-GRU in software defined networks. Various malicious users tend to attack the cognitive radio network [5] during cooperative spectrum sensing.

The authors of [6,7] explained spectrum sensing, which takes place in cognitive radio networks [8]. In addition, they put forth recent advances and challenges that have 1occured in CRN. Ref. [9] surveys the attacks [10] and their detective measures in the physical layer of CRN. These attacks are mitigated using machine learning [11] and deep learning algorithms [12,13], as mentioned in [14,15]. Both the fusion centre and the secondary users have been trained with the typical activities of legitimate primary users in RNN during the early stage of spectrum sensing [16]. The secondary users (SUs) analyse the behaviour of the incoming primary signal using the training data when the licensed channel is shared with the secondary users.

There are no malicious behaviours in any of the training data [17]. As a result, in a real-world setting, the RNN cannot recognise and predict malevolent users. However, by mapping the input series of length *l* to the label domain of length $2^{l}$, we may construct the RNN to detect and anticipate the threat.

After mapping, we compared the anticipated likelihoods of each label with the actual received series, then calculated the mean square error to determine the loss. However, because of the problem of gradient vanishing, RNN cannot be employed for long-term reliance. RNNs are prone to forgetting long-term data because of the gradient vanishing problem. Therefore, they are unable to utilise patterns that occur before the current input series.

There have been suggestions from different recurrent network topologies to overcome this drawback. The long short-term memory (LSTM), which swaps out the usual hidden layer node for a memory cell and several gates, is one of the major RNN improvements.

In terms of classification accuracy, GRU [18,19] performs just as well as LSTM. Due to the fact that it [20] uses fewer tensor operations than LSTM, training is discovered to be faster in GRU [21,22]. 1D-CNN was employed by the authors in [23] to distinguish between malicious and authorised users. The comparative performance of CNN, RNN, LSTM, DNN, and GRU may be found in [24].

To reduce the overall service time of spectrum handoff to the ideal value, particle swarm optimization is employed in [25]. According to numerical findings, the authors made the method dramatically reduced to total service time when compared to other spectrum handoff strategies.

A system of fuzzy logic-based spectrum handoff has been introduced in [26]. The system comprises of a fuzzy logic controller (FLC) with inputs such as eNB load, received power, arrival rate of primary users. The suggested method offers a strong framework for preventing unwanted CRN handoffs in LTE-A systems [27].

In this paper, we used energy detection method for spectrum sensing and the results of those sensing are used to train GRU and SVM classifier. We have added SVM classifier in addition to the GRU for better classification and employed in CRN, which was not discussed in any of the literature. The training phase is followed by testing to differentiate and segregate the authenticated and malicious users in the cognitive radio network. The novelty of this paper is to use the combined models of GRU and SVM in CRN.

The paper is sectioned as: Section 2—Related Work, Section 3—Key Terminologies and Techniques, Section 4—Proposed Methodology with System Model, Threat Detection and Dataset Preprocessing, Section 5—Analysis of Computational Complexity, Section 6—Results and Discussion, and Section 7—Conclusions.

## 2. Related Work

Primary user emulation attack is detected and prevented in [28] using extreme machine learning method and time–distance with the signal strength evaluation method. The former method is used to give decisions on malicious attacks and the latter method identifies the malicious users efficiently by bringing down overall sytem delay, increasing system utilization and performance of the network.

In [29], a primary user emulation attack has been dealt in a cognitive radio network using a genetic artificial bee colony and the detection probability has been enhanced. The authors used two main thresholds predefined to indicate primary user presence and primary user emulation attack presence. They considered four parameters of cognitive radio.

Primary user emulation attack detection in CRN using multi layer LSTM model achieves superior performance compared to single layer LSTM in [30]. In order to ensure that recent forecasts are influenced by long-term data, LSTM is implemented using gates, which control the rate of learning and forgetting.

In [31], a new iteration of the LSTM, the GRU with fewer gates and the ability to preserve long-term reliance, has been introduced with SVM to detect the intrusion in the data of network traffic. The authors compared GRU-SVM model and GRU–Softmax model to prove that GRU performs well with SVM classifier in terms of training and testing accuracy.

Distributed denial of service cyber attacks have been identified using GRU, Naive Bayes, RNN and sequential minimal optimization models. The authors evaluated better accuracy, recall, F1-score, and precision to prove its efficiency in [32].

In order to improve network performance by limiting attacks, a Hierarchical Cat and Mouse Based Ensemble Extreme Learning Machine model has been suggested in [33] to analyse security concerns in CRN. Additionally, an attempt is made to determine whether machine learning classification can successfully identify SSDF attacks in a binary reporting CRN. The temporal properties of SU are acquired using the history of the sensing reports accumulated at the FC, establishing the training and testing data-sets.

In [34], malicious signal identification using machine learning is applied to cognitive radio networks. Riverbed simulation software simulates the design of cognitive radio users and the network environment. Whether the received signal is a secure detecting signal or a malicious signal, it is managed. For the purpose of classifying spectrum sensing signals as malicious, suspicious, or secure signals, a fuzzy logic-based system is used. Fuzzy logic parameters are extracted from the top three machine learning characteristics out of a total of 49 features. When compared to other schemes in the literature, the security of primary users is increased.

In full-duplex cognitive radio networks, the authors of [35] offer a robust cooperative spectrum sensing framework based on ensemble machine learning (EML) that is accurate and resilient against malicious interference and attacks. By enabling secondary users (SUs) to sense and transmit simultaneously over the same frequency range, FD communication increases their capacity for spectrum awareness. An EML framework has been created that offers reliable and accurate fusion performance to lessen the impact of interference and attacks.

The authors of [36] provide a framework with two layers for categorising Byzantine attackers in a CRN. In the first layer, called Processing, the Hidden Markov Model is used to obtain a probabilistic link between the states of the PU and the sensing reports of the SUs. This creates the necessary dataset for the following layer. The second layer, the Decision layer, divides the SUs into Byzantine attackers and typical SUs using a variety of ML methods. The learning classifiers deliver good performance across a range of testing parameters, according to extensive simulation findings. The method used in [36] is more robust, especially when there are many malicious users present, according to a comparison examination of it with an existing non-ML strategy.

In [37], the authors promote the application of reinforcement learning to obtain optimal or nearly ideal security enhancement solutions by identifying different hostile nodes and their attacks in CRNs. Reinforcement learning, an artificial intelligence approach, has the capacity to both learn new assaults and recognise those that have already been learnt.

By examining the transmitter’s received power, [38] suggests an adaptive learning-based assault detection in CRN for spotting and preventing PUEA. By separating a low spectrum legal PU from an attacker, the learning process supports various advantageous aspects. Cyclostationary feature analysis is used in the learning method to identify adversaries and low power PU in CR communications.

Jamming attack and primary user emulation attack have been identified and detected using one dimensional CNN [23] in cognitive radio networks and the results were validated in terms of quality metrics and numerical simulation results.

## 3. Key Terminologies

### 3.1. Cognitive Radio Network

A cognitive radio is an intelligently programmed radio that uses the best channel available to prevent interference and congestion. Primary or licensed users and secondary or unlicensed users are the two different categories of network users. For the benefit of radio spectrum users, secondary users take over available channels when they sense them in order to enhance service quality. Due to the ineffective utilisation of licensed spectrum, some frequency bands were overloaded while others were underutilised. This ineffective spectrum use, which affects service quality, can be addressed through cognitive radio.

The use of cognitive radio networks has been increasing for quite an amount of time due to some of its distinguishing characteristics such as self-organization ability and cognitive flexibility and reconfigurability. These attributes have helped the CRN to grow rapidly.

### 3.2. Spectrum Sensing

One of the most important components of cognitive radio networks is spectrum sensing. A cognitive radio can learn about its surroundings and spectrum availability by spectrum sensing. Various types of spectrum sensing are distributed, centralized, and hybrid.

### 3.3. Energy Detection

Energy detection is a sort of sensing that has cheap application costs and no prerequisite knowledge requirements for the primary user. To assess if the channel is open, each secondary user’s detector compares the sensing threshold with the signal energy detected. If the sensing threshold is greater than the measured energy, the hypothesis $H_{0}$ is adopted. If the sensing threshold is less than the energy being measured, hypothesis $H_{1}$ is adopted. If there are no active main users present in the channel, the cognitive user sends the fusion centre a one-bit hard local decision of “0.” If the cognitive user detects an active main user, it sets the local decision to “1” and sends it to the fusion centre.

The fusion centre formulates a collaborative global decision on the primary user’s activities by obtaining precise local assessments from the cognitive users. It is also possible to make soft decisions, but this requires a significant amount of processing complexity and bandwidth. Thus, hard decisions are usually preferred.

### 3.4. Gated Recurrent Unit

The improved version of traditional recurrent neural network is GRU. The vanishing gradient problem that bedevils a typical RNN is addressed by GRU using two gates, the update gate and the reset gate. These two vectors essentially decide what information should be transmitted to the output. GRUs are smart enough to keep hold of useful past information and delete the unwanted information. It consists of two gates: (i) update gate (ii) reset gate.

#### 3.4.1. Update Gate

The update gate is similar to the output gate of LSTM. It is responsible for how much old information should be retained and sent to the next stage.

#### 3.4.2. Reset Gate

The reset gate is similar to the combined effect of the input gate and the forget gate of LSTM. It is responsible for how much old information to forget.

### 3.5. Support Vector Machine

Support Vector Machine or SVM, is a technique that is utilised for tasks involving both classification and regression problems. However, it is widely used to solve classification problems. The support vector machine is more popular since it delivers remarkable accuracy while consuming less processing power. The major goal of SVM is to classify the datapoints in the N-dimensional space of a hyperplane.

SVM is highly useful when the data are not known very well. It can be applied to data that are irregularly dispersed and whose distribution is unknown. It also supports high dimensional data that it can be used in machine learning field. It can be used for image data, text data, audio data, and much more. We may handle any complex problem by using the related kernel function, which is a very helpful strategy offered by the SVM. Kernel offers the option of selecting the function that is not always linear and that can take on different forms depending on the kind of data it uses, making it a non-parametric function.

SVM often does not experience over fitting and performs well when there is strong evidence of class separation. When the total number of samples is fewer than the number of dimensions, SVM can be utilized and it performs well in terms of memory. It has good performance in N-dimensional space.

The SVM gives an effective and unique solution by optimizing the various solutions of samples corresponding to each local minimum. It is robust even when the training sample contains some bias. Support vector machine predicts faster with better accuracy and good computational complexity.

## 4. Proposed Work

In this section, we introduce a system model and a proposed methodology.

### 4.1. System Model

A licensed user, *S* secondary users, and *M* malevolent users make up the proposed network system, where M<S. In order to determine the signal status of the licensed user, all *S* secondary users and *M* malevolent users engage in a spectrum sensing. The transmitted licensed user signal is given by
(1)L=Bv
where $v={\left[v_{1},v_{2},v_{3},\cdots ,v_{K}\right]}^{T}$ denotes the transmitted data vector, with *K* representing the Kth sampling time, BK×K is a coefficient matrix with $k_{ij}\;;i=j=1,2,3,\cdots ,K$ describing the channel coefficients. The samples as mentioned as the elements of v are given by
$$v_{i}\left(t\right)=\sum_{l=-\infty }^{\infty }d_{l}p\left(t-lT_{s}\right)e^{j2\pi f_{c}t}$$
where *d* denotes digitally modulated data symbols, p(t) denotes pulse shaping filter, $f_{c}$ represents carrier frequency, and $T_{s}$ denotes symbol duration.

The received signal at the $s^{\mathrm{th}}$ secondary user is given by
(2)$$g_{s}\left(t\right)=\left\{\begin{array}{cc} n_{s}\left(t\right) & H_{0}\\ h_{s}L\left(t\right)+n_{s}\left(t\right) & H_{1} \end{array}\right.$$
where $g_{s}\left(t\right)$ denotes signal received at the $s^{\mathrm{th}}$ secondary user, L(t) denotes transmitted licensed user’s signal, $n_{s}\left(t\right)$ is the additive white gaussian noise and $h_{s}$ represents channel gain between licensed user and secondary user. In addition, the hypotheses $H_{0}$ and $H_{1}$ in (Equation 2) represent the absence and presence of licensed user, respectively.

Each secondary user detects the primary user’s status and transmits local decisions to the GRU model to separate the malicious user from the trustworthy users. Then, using the “AND” rule, the fusion centre finalises the global decision by analysing just the authorized local decisions. The proposed methodology uses the energy detection method to sense the spectrum because it does not demand for any prior knowledge of the primary user. Energy of the received signal is assumed to be the test statistics and the same is given by,
(3)$$E_{s}\left(i\right)=\left\{\begin{array}{cc} \sum_{t=t_{i}}^{t_{i}+U-1}{\left|n_{s}\left(t\right)\right|}^{2} & :H_{0}\\ \sum_{t=t_{i}}^{t_{i}+U-1}{\left|hL\left(t\right)+n_{s}\left(t\right)\right|}^{2} & :H_{1} \end{array}\right.$$

In (Equation 3), *U* number of samples are considered. As number of samples in (Equation 3) are more, we assume that $E_{s}$ in (Equation 3) follows gaussian density as given by,
(4)$$E_{s}∼\left\{\begin{array}{cc} N\left(U,2U\right) & :H_{0}\\ N\left(U\left(\gamma_{s}+1\right),2U\left(\gamma_{s}+1\right)\right) & :H_{1} \end{array}\right.$$
where $N\left(\mu ,\sigma^{2}\right)$ denotes normal density function with mean μ and variance $\sigma^{2}$, and $\gamma_{s}$ represents signal-to-noise ratio between licensed user and $s^{\mathrm{th}}$ secondary user.

Each secondary user detects the primary user’s status and transmits local decisions to the GRU model to separate the malicious user from the trustworthy users. Then, using the AND rule, the fusion centre finalises the global decision by analysing just the authorized local decisions.

Let $x_{s}^{<t>}$ denote the local decisions of user *s* at a time instant *t*. Then, the local decision is given by
(5)$$x_{s}^{<t>}=\left\{\begin{array}{cc} 1 & ;E_{s}\geq \lambda_{s}\\ 0 & ;E_{s}<\lambda_{s} \end{array}\right.$$

### 4.2. GRU-SVM Based Threat Detection

Our suggested approach employs GRU to identify malevolent users who interfere with the cognitive radio network. This model is one that holds promise for time-series data because it takes old records on predictions into account. Figure 1 depicts the architecture of a GRU.

In the architecture shown in Figure 1, $x^{<t>}=\left[\begin{array}{c} x_{1}^{<t>}\\ x_{2}^{<t>}\\ \vdots \\ x_{s}^{<t>} \end{array}\right]$ represents the stacked arrangement of local decisions received from all secondary users at time *t*. The network shown in Figure 1 is earlier trained using the training vector of similar characteristic.

Different neural networks are utilised as the gates in GRU, which determine whether information is retained or forgotten. The update gate and reset gate are two separate gates that are used to do this. Figure 2 depicts the update gate, which controls how much historical data (memory) should be kept. Figure 3 depicts the reset gate, which determines how much of the prior memory to erase. The output is normalised from 0 to 1 using the sigmoid activation function, which is used by the update gate and reset gate. Therefore, if any value is multiplied by “0”, the result is “zero”, and the information is lost. Any value multiplied by “1” yields “one”, indicating that the information was maintained. $x^{<t>}$ is the input data, $h^{<t-1>}$ is the hidden state at time $‘t-1^{\prime }$, and $h^{<t>}$ is the hidden state at the output at time ‘t’. Using the weight matrices $U_{r}$, $W_{r}$ and bias vector $b_{r}$ of the neural network of the reset gate, the reset gate concatenates $h^{<t-1>}$ and $x^{<t>}$. The resulting vector is activated using a sigmoid function to produce $r^{<t>}$, the normalised output between 0 and 1.
(6)$$r^{<t>}=\sigma \left(W_{r}x^{<t>}+U_{r}h^{<t-1>}+b_{r}\right)$$

The weight matrices $U_{z}$, $W_{z}$ and bias vector $b_{z}$ of the neural network of the update gate are used to combine $h^{<t-1>}$ and $x^{<t>}$. The resulting vector is activated using the sigmoid function to produce $z^{<t>}$, normalised output between 0 and 1.
(7)$$z^{<t>}=\sigma \left(W_{z}x^{<t>}+U_{z}h^{<t-1>}+b_{z}\right)$$

In order to create the hidden state $h^{<t>}$ at the present time ‘t’ and the output $y^{<t>}$, $z^{<t>}$ and $r^{<t>}$ are used. Point-wise multiplication is performed between the previous hidden state $h^{<t-1>}$ and the output of the reset gate, $r^{<t>}$. It is combined with the input vector $x^{<t>}$ to produce the output. In order to control neural network output and avoid excessive or undersized data between iterations, the concatenated output is given to the hyperbolic tangent activation function, or tanh, which normalises the output between −1 and 1. The output of this neural network ${\tilde{h}}^{<t>}$ is given by,
(8)$${\tilde{h}}^{<t>}=\tanh\left(W_{h}x^{<t>}+U_{h}\left(r^{<t>}⊙h^{<t-1>}\right)+b_{h}\right)$$
where the neural network’s weight matrix and bias vector, are $W_{h}$ and $b_{h}$, respectively. The ${\tilde{h}}^{<t>}$ output of the neural network determines which data to discard and which new data to add. By performing point-wise multiplication between $1-z^{<t>}$ and $h^{<t-1>}$, thrown off is conducted. Multiplying $z^{<t>}$ and ${\tilde{h}}^{<t>}$ is the process of adding additional data. The hidden state $h^{<t>}$, is then produced by point-wise addition using these two functions of adding new data and discarding data.
(9)$$h^{<t>}=z^{<t>}⊙{\tilde{h}}^{<t>}+\left(1-z^{<t>}\right)⊙h^{<t-1>}$$

The above-described procedure pertains to a single GRU. To increase accuracy, we can increase the number of GRU cells in the GRU layer. Figure 4 displays the proposed model’s neural network representation. The input layer ($x_{1},x_{2},\cdots x_{s}$) is the first layer, and *s* is the number of local decisions taken from the spectrum sensing results. Then, it is transferred to the GRU layer ($G_{1},G_{2},\cdots G_{n}$) with *n* number of neurons (cells), which is followed by the drop-out layer with a 0.5 drop-out rate. This layer aids in addressing the overfitting issue. The fully connected layer, which is the following layer, uses *d* number of neurons to conduct global model classification. The SVM classifier then uses a single neuron $C_{1}$ in the final output layer to classify the trustworthy users and the malevolent users.

There are *s* inputs given to the proposed neural network model. Parameters are learnt through Equations (Equation 6)– (Equation 9) of GRU gating mechanism. The parameters can also be learnt by optimizing L2-SVM, which is more stable and differentiable, whose function is shown below
(10)$$\min\frac{1}{2}{∥q∥}_{2}^{2}+C\sum_{i=1}^{d}\max{\left(0,1-\delta \left(q^{T}f+\beta \right)\right)}^{2}$$
where, $f={\left[f_{1}f_{2}\cdots ,f_{d}\right]}^{T}$, $q={\left[q_{1},q_{2}\cdots ,q_{d}\right]}^{T}$ is vector for linear combination in SVM and β is constant. The softness of margin is determined by a parameter of the problem named *C*. When *C* is a positive number, it is a hard SVM; if *C* is zero, the SVM will just choose q=0 and cares nothing for accuracy. In reality, a few *C* values are used to test how the SVM should function.

For predicting the score vector of each class, $g\left(f\right)=\mathrm{sgn}\left(q^{T}f+\beta \right)$ decision function is used. Predicted class label *y* of data *x* is obtained by using *argmax* function. The highest score in the vector of predicted classes will be returned by Equation (Equation 11) as its index.
(11)$$\mathrm{Predicted}\;\mathrm{class}=\mathrm{argmax}\left(\mathrm{sgn}(q^{T}f+\beta )\right)$$

The summarized flowchart of GRU-SVM model is given in Figure 5. Generally, Softmax layer is used at the final layer for classification. In this paper, we are using SVM as it has the following advantages over Softmax layer. The Softmax layer is suitable for multinomial classification, while SVM goes well for binary classification. SVM simply needs its margins to be satisfied and is not concerned with the specific scores of the classes it predicts. In contrast, by guaranteeing that the correct class has the higher/highest probability and the erroneous classes have the lower chance. The Softmax function will constantly find a method to enhance its predicted probability distribution. Although the Softmax function’s behaviour is excellent, it is overkill for a binary classification issue. As we know, GRU solves the problem of the vanishing gradient. However, as a result of this, the Softmax layer, when it is added to the GRU model, misclassifies data. The GRU–Softmax model’s aforementioned error actually works towards the GRU-SVM model’s benefit. The practicality of selecting SVM over Softmax in this instance, however, is not just based on a comparison of the demonstrated prediction accuracies of the two models. The length of the testing and training periods was also taken into account. SVM outperforms Softmax, as suggested by the computational complexity of each of them.

### 4.3. Dataset Preprocessing

In our paper, we used 16.89 MB csv file dataset which is available at kaggle [39]. A total of 80% of the dataset is used for training and 20% is used for testing.

The dataset is discretized after initially being normalised. Standardization performs continuous data normalization. StandardScaler().fit_transform() function of Scikitlearn for efficiency is used to perform data standardization. Indexing performs for categorical data normalization. LabelEncoder().fit_transform() function of Scikit-learn is used for mapping the categories to [0,n−1]. The bin number is then assigned in accordance with the indices after the continuous features are discretized using the quantile of the features. The classification performance of the datasets is improved through discretization or binning, which also lowers the processing cost. The features are then used to execute one hot encoding.

## 5. Analysis of Computational Complexity

In this section, we present the computational complexity of proposed method. It is assumed that sensing is implemented after every 2 s, followed by local decision and GRU implementation. Therefore, the computational complexity shown here is applicable once after 2 s. As shown in Equation (Equation 3), energy detection requires 2U number of multiplications and 2U−1 number of additions. Therefore, the complexity of energy detection is given by O(2U). Whereas when GRU is considered, as per Equations (Equation 6) and (Equation 7), the computational complexity of update gate stage and reset gate stage is O(2S) additions and multiplications and O(2S) additions and multiplications, respectively. Moreover, considering the two sigmoid operations, it requiresrequire two additions, two divisions and two exponents. The computational complexity of Equation (Equation 8) is O(3S) multiplications and O(S+1) additions, respectively. The computational complexity of Equation (Equation 9) is O(3S) multiplications and O(S+1) additions, respectively. Therefore, the total computation complexity of the proposed method is O(2U+4S+1) number of additions and O(2U+8S) number of multiplications, which is linearly increasing as a function of *U* and *S*. Usually *S* and *U* are of the order of not more than a few hundred. Therefore, the complexity of the algorithm is very less.

## 6. Results and Discussion

The simulations were run on a 2nd generation Intel Xeon Scalable Processor with up to 28 cores per processor, a Tesla 300 GPU, up to 16 × 2.5″ SAS/SATA/SSD storage with a maximum capacity of 122.88 TB, and 24 DDR4 DIMM slots for memory modules (12NVDIMM or 12 DCPMM only). Using regular classification metrics such as accuracy, precision, sensitivity (recall), F1 score and specificity, the effectiveness of the different models, including DNN, CNN, LSTM, LR, and KNN was compared with the proposed method. For the study, a DNN with three hidden layers—100, 40, and 10 neurons—was built. The size of the kernel for CNN was 16, 8, and 3, respectively, with three layers of 64, 32, and 16 filters.

The implementation of the LSTM and GRU-SVM models uses the same number of cells, n = 32. Three neighbours were used in the implementation of the KNN algorithm. Finally, default Sklearn settings were applied to LR. The majority of the approaches were tested across 100 epochs because they converge with relatively few iterations.

Different classification metrics are shown in Figure 6 specifically, accuracy, precision, sensitivity, F1 score, and specificity when utilising our proposed model to train and evaluate the data. Better precision, specificity, accuracy, and F1-measure were found through testing. Table 1 lists the values of the assessment metrics that were determined after training and testing the data.

In Figure 7, the training accuracy of several approaches including CNN, DNN, LSTM, GRU-SVM, LR, and KNN is plotted across various number of epochs. When training the data, our suggested model excels in accuracy. Training accuracy of the proposed work attains 80.99% which is comparatively higher than the compared methods. Additionally, LSTM and CNN appear to be consistent with GRU-SVM’s.

In Figure 8, the testing accuracy of different approaches is displayed throughout the number of epochs, and the proposed model outperforms other strategies in terms of testing data accuracy. Training accuracy of the proposed work is 82.45% which is comparatively higher than the compared methods. LSTM performs quite similarly to GRU-SVM, demonstrating the effectiveness of RNN variants for assessing time series data. The probability of detection is plotted against the number of epochs in Figure 9. The proposed model reaches probability 1 at 65 epochs whereas all the other compared models reach detection probability after 75 epochs. Our suggested model excels at detection across a range of epochs.

The probability of false alarm is plotted against the number of epochs in Figure 10, where the proposed model performs well since it has the lowest false alarm probability across all attempted epochs. At 97 epochs, the false alarm probability got reduced to 0 (reaches minimum) compared to other compared methods, which shows better performance.

The confusion matrix for the training phase and testing phase for the normal case and under attack case is explained in Figure 11 and Figure 12. TheConfusion Matrix is a performance measurement for two class classification. It is a table with four separate sets of actual and predicted values. In our paper, we classify authorized and malicious ones.

The created Confusion Matrix in Figure 11 and Figure 12 have four separate quadrants: Top-Left Quadrant → True Negative, Top-Right Quadrant → False Positive, Bottom-Left Quadrant → False Negative, Bottom-Right Quadrant → True Positive. False denotes a mistake or incorrect prediction, while True denotes that the values were correctly predicted.

Let us calculate different measures to quantify the model quality.


**Precision:**
Precision is the percentage of truely positives, of all the predicted positives.
$$\mathrm{Precision}\;\left(\mathrm{training}\right)=\frac{\mathrm{True}\;\mathrm{positive}}{\mathrm{True}\;\mathrm{positive}+\mathrm{False}\;\mathrm{positive}}=79.0306\%$$
Precision(testing)=89.3005%
**Sensitivity (Recall):**
The model’s sensitivity indicates how well it can predict positive outcomes. Sensitivity is the percentage of predicted positives, of all the positive cases.
$$\mathrm{Sensitivity}\;\left(\mathrm{training}\right)=\frac{\mathrm{True}\;\mathrm{positive}}{\mathrm{True}\;\mathrm{positive}+\mathrm{False}\;\mathrm{Negative}}=84.3726\%$$
Sensitivity(testing)=78.5418%
**Specificity:**
The model’s specificity indicates how well it can predict negative outcomes.
$$\mathrm{Specificity}\;\left(\mathrm{training}\right)=\frac{\mathrm{True}\;\mathrm{Negative}}{\mathrm{False}\;\mathrm{positive}+\mathrm{True}\;\mathrm{Negative}}=77.6132\%$$
Specificity(testing)=87.6022%
**F1-score:**
The “harmonic mean” of sensitivity and precision is called the F1-score. It is suitable for imbalanced datasets and takes into account both false positive and false negative cases. The True Negative values are not considered in this score:
$$F1\text{-score}\;\left(\mathrm{training}\right)=\frac{2*(\mathrm{Precision}*\mathrm{Sensitivity})}{(\mathrm{Precision}+\mathrm{Sensitivity})}=81.6142\%$$
F1-score(testing)=83.5763%
**Accuracy:**
The model’s accuracy indicates how frequently it is accurate.
$$\mathrm{Accuracy}\;\left(\mathrm{training}\right)=\frac{\mathrm{True}\;\mathrm{positive}+\;\mathrm{True}\;\mathrm{Negative}}{\mathrm{Total}\;\mathrm{predictions}}=80.9929\%$$
Accuracy(testing)=82.4515%

The proposed model distinguishes malicious users from genuine users effectively, outperforming all other deep learning and machine learning models in terms of metrics parameters, as shown by the results.

## 7. Conclusions

In this study, we used GRU to identify malevolent users while sensing the spectrum. Results show that this model is effective at classifying fraudulent users from legal ones when combined with an SVM classifier, with testing accuracy of 82.45 percent and training accuracy of 80.99 percent. The probability of detection achieves 1 at 65 epochs. Additionally, we have contrasted our proposed model with other deep learning and machine learning algorithms, such as CNN, LSTM, LR, DNN, and KNN. In terms of training accuracy, testing accuracy, detection probability, and false alarm probability, GRU-SVM performs better than traditional models.

## Figures and Tables

**Figure 1 sensors-23-01326-f001:**
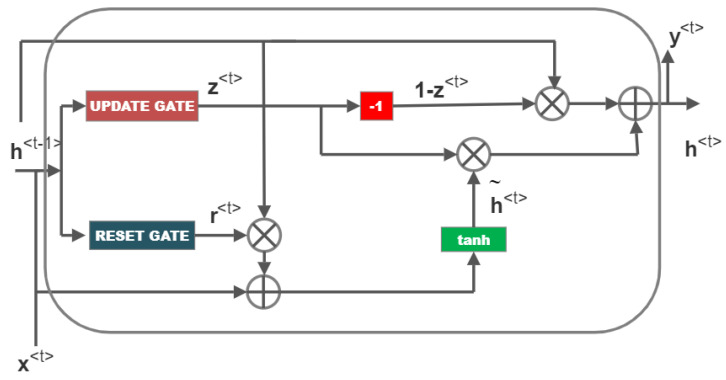
GRU cell.

**Figure 2 sensors-23-01326-f002:**
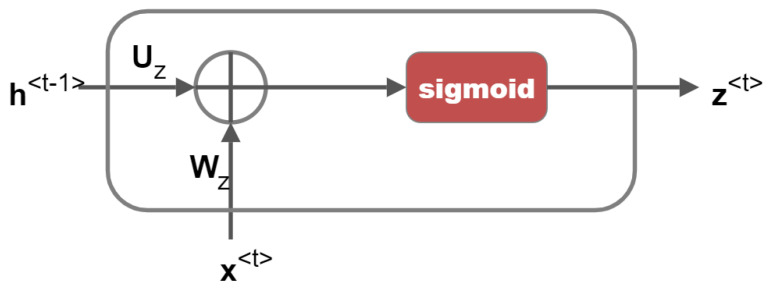
Update Gate.

**Figure 3 sensors-23-01326-f003:**
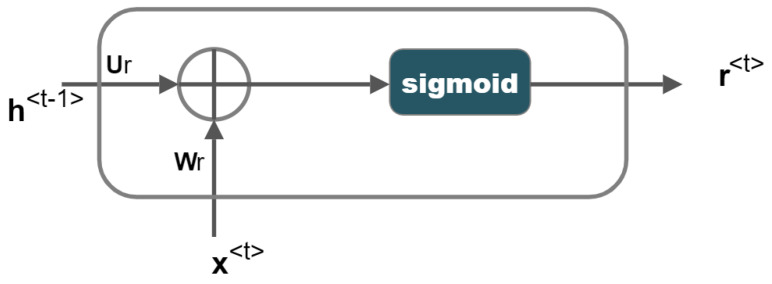
Reset Gate.

**Figure 4 sensors-23-01326-f004:**
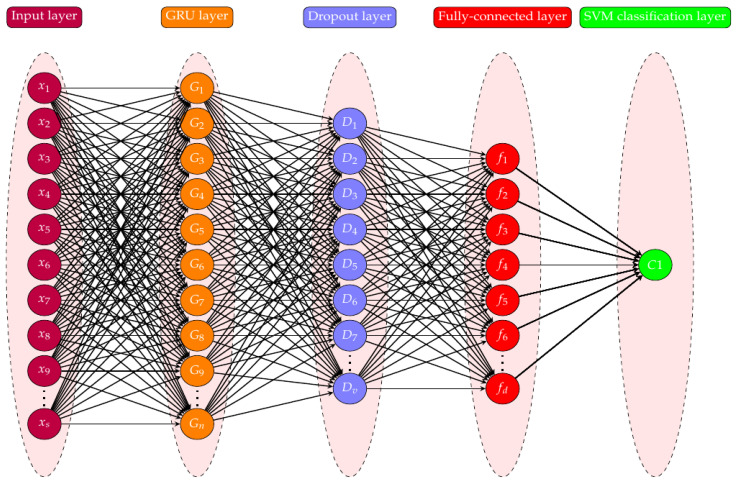
Neural network representation of proposed GRU-SVM model.

**Figure 5 sensors-23-01326-f005:**
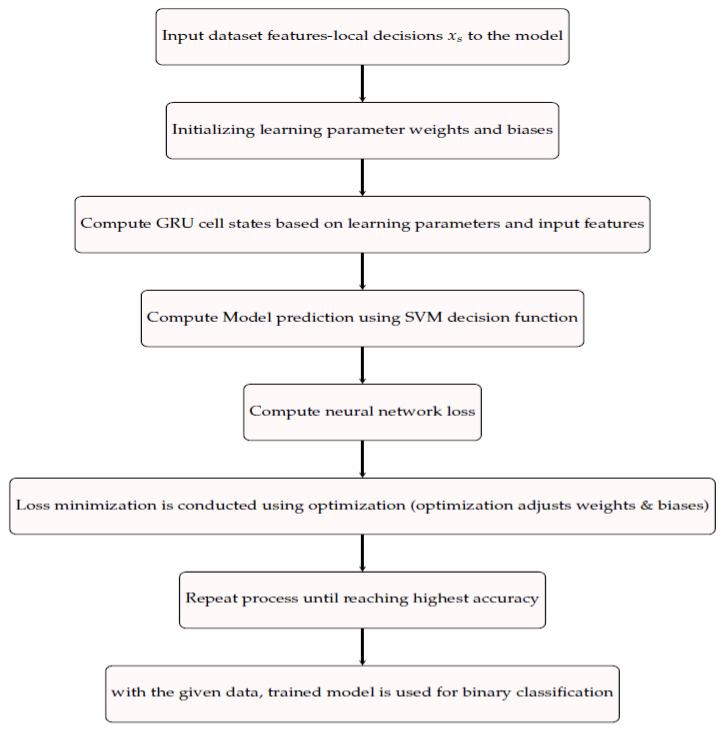
Flowchart for classification of malicious and legitimate users using GRU-SVM model.

**Figure 6 sensors-23-01326-f006:**
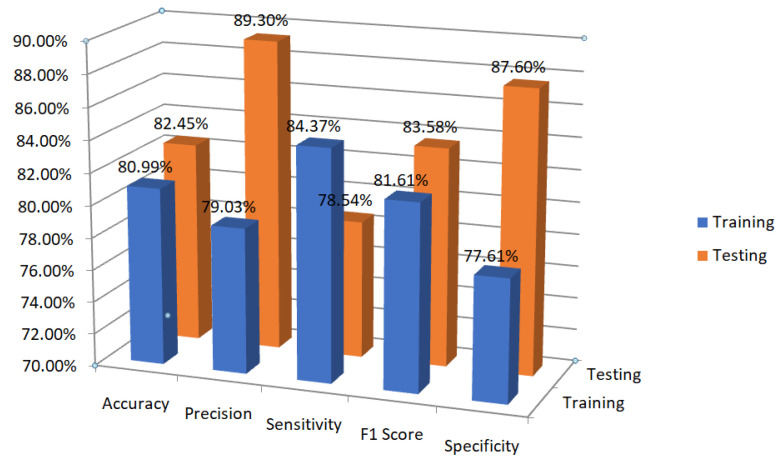
Evaluation metrics chart for training and testing data using GRU-SVM.

**Figure 7 sensors-23-01326-f007:**
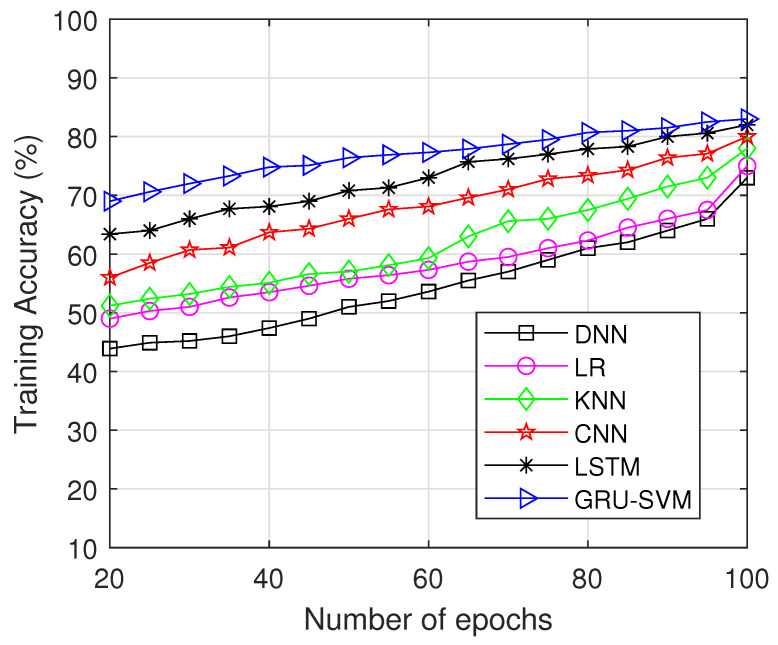
Training accuracy vs. Number of Epochs.

**Figure 8 sensors-23-01326-f008:**
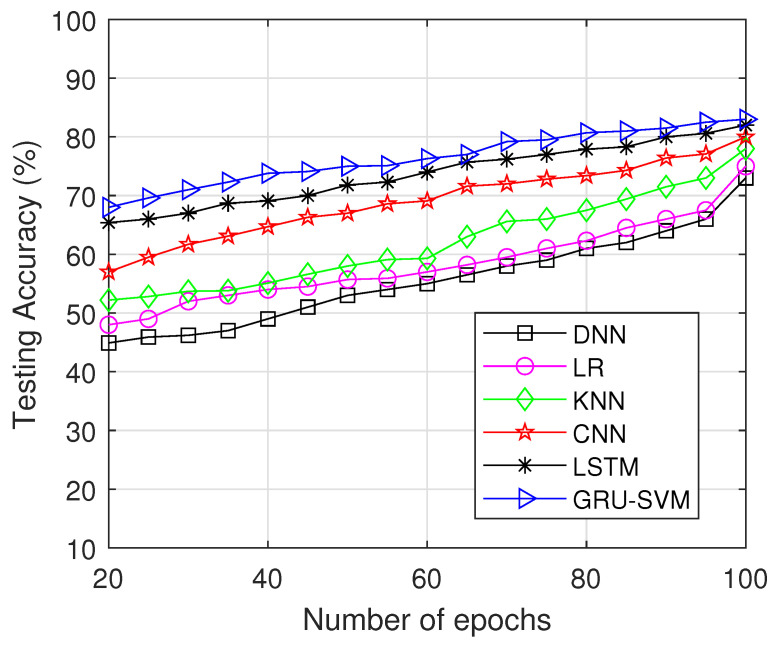
Testingaccuracy vs. Number of Epochs.

**Figure 9 sensors-23-01326-f009:**
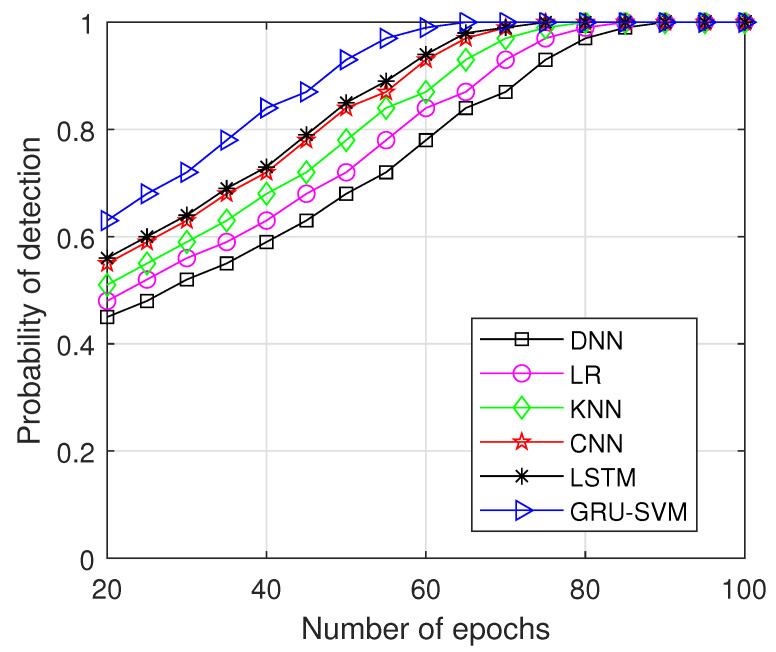
Probabilityof detection vs. Number of Epochs.

**Figure 10 sensors-23-01326-f010:**
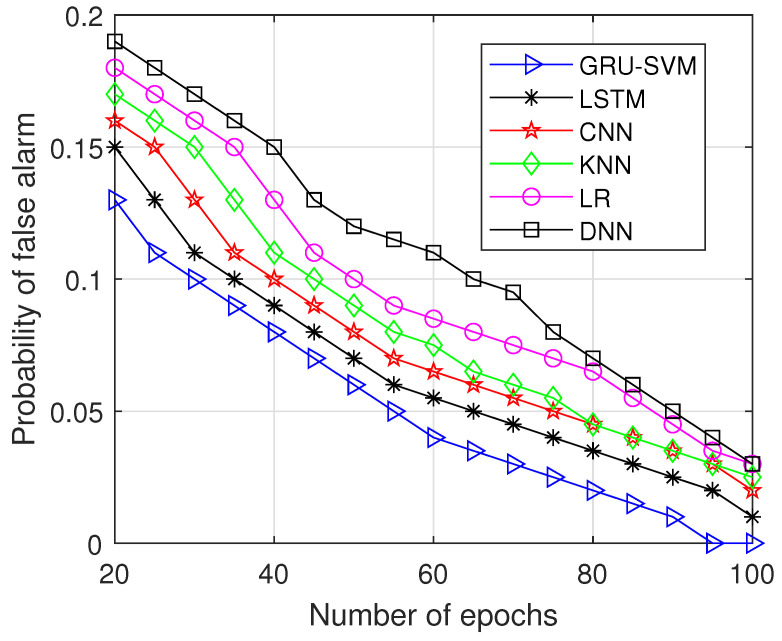
Probability of false alarm vs. Number of Epochs.

**Figure 11 sensors-23-01326-f011:**
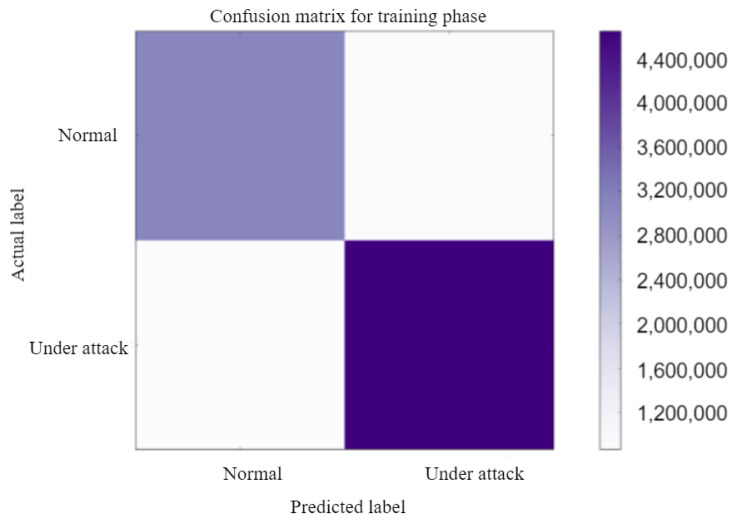
Confusion matrix-training.

**Figure 12 sensors-23-01326-f012:**
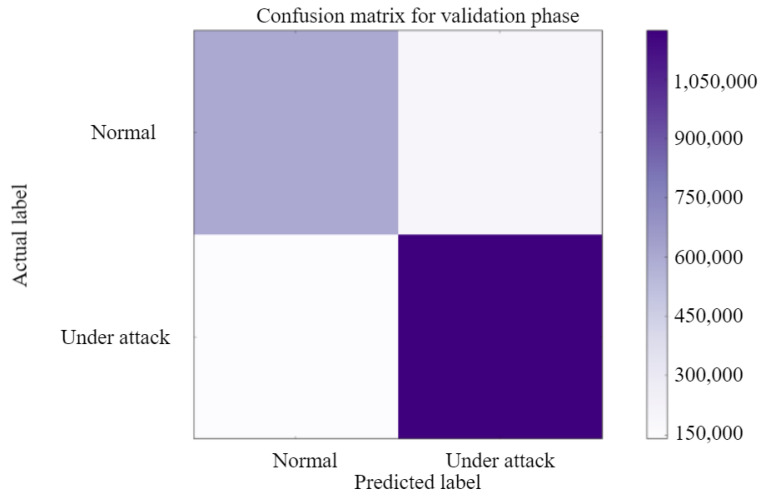
Confusion matrix-validation.

**Table 1 sensors-23-01326-t001:** Evaluation metrics values.

Evaluation Metrics	Training	Testing
Accuracy	80.9929	82.4515
Precision	79.0306	89.3005
Sensitivity (Recall)	84.3726	78.5418
F1 Score	81.6142	83.5763
Specificity	77.6132	87.6022

## Data Availability

Not applicable.

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
