# Peer review of "GRU-SVM Based Threat Detection in Cognitive Radio Network"

_sensors, 2023, doi:10.3390/s23031326_

Round 1

Reviewer 1 Report

This paper presents the used GRU and SVM to distinguish between authorised users and malicious users in cognitive radio network. The overall logic of the article is clear and well expressed. But I still have the following questions and suggestions to improve the quality of the article.

1)    This article places SVM behind GRU as a classifier and has achieved good results, but the author does not explain why to use SVM in this article. Is there theoretical support? Or is SVM better as a classifier after experimental comparison? Please add a discussion or experiment.

2)    Please consider whether you should modify the topic to add SVM to the topic.

3)    It is recommended to add GRU-SVM method to the paper to compare its efficiency with other methods.

4)    Please check the formula content carefully, such as formula 4.

5)    Please improve the overall image quality. It is recommended to use vector images.

Reviewer 2 Report

A novel approach for threat detection in CRs is addressed, specifically with the combination of SVM and GRU for improving the tranining accuracy.  The reviewer thinks the work is interesting and as well complete, since it provides some insightful simulations. The suggestion is that:

1) provide the analysis of the computational complexity of the proposed approach.

2) make a clear statement that the input of the signal for the model training, and as well the test signal set.

Reviewer 3 Report

The authors employed a Gated Recurrent Unit (GRU) to identify malevolent users who interfere with the cognitive radio network. The paper has limited contributions and still needs more improvements. I have some comments and suggestions as follows:

- Abstract lines 6 -9, the sentences are not clear. Please rewrite those sentences and describe your novelty and short result, like overall accuracy. 

- in the introduction, the citations are properly cited, starting with citation number 7. Please check all the citiation oerderong. The paper's contributions are not clearly described. Please rewrite your papers' contributions; you may break them down into several points. Also, the paper's organization section numbering should be in Arabic numbers, not Roman numbers.

- The authors should add a new section for related work because this paper lacks references. You must add works that discuss the cognitive radio network, and you may add the following works:  Analysis of spectrum handoff schemes in cognitive radio network using particle swarm optimization",  "Efficient handoff spectrum scheme using fuzzy decision making in cognitive radio system" and  "An intelligent spectrum handoff scheme based on multiple attribute decision making for LTE-A network" 

- Section 2 is inadequate and needs more details

- The definition term " Gated Recurrent Unit (GRU)" is repeated several times throughout the paper. 

- In subsection 3.3. Dataset preprocessing, and dataset description is missing, such as size, type,..etc. Is the dataset available online?  How do you divide the dataset for training and testing? 

- Please improve the quality of Figures 7 - 9 and add more discussion. Also, on page 7, line 157, "Better precision, specificity, accuracy, and F-measure were found through testing. 1 lists the values".  There is a missing word, " Table," before number 1. 

- The final results, such as accuracy and detection percentage, should be given in the conclusion.

- KNN is missing the abbreviation list. 

Round 2

Reviewer 1 Report

The authors have improved the manuscript according to the comments. Then, I have no more questions and I suggest to accept the manuscript.

Reviewer 3 Report

Thanks for addressing some of my comments, but still there a few concerns such as:

- You should not begin the stance with a citation, like [6] explains the spectrum sensing, you should use for example "In [] proposed .... or The authors of [] proposed.... "

- The paper's contribution is not clearly stated. you should discuss your paper contribution before the paper's organization.

- The related work section is still short, you may more details. 

- The available dataset should be cited not putting the link in the text. 

- A most important point, is that figures 7 -10 should be moved within section 6. Please address this point properly. 

- Please check all grammatical errors throughout the paper. 
